# An Attention-Based 3D Convolutional Autoencoder for Few-Shot Hyperspectral Unmixing and Classification

**Chunyu Li** [1,2,3], **Rong Cai** [1,2,*] **and Junchuan Yu** [4]

1    Aerospace Information Research Institute, Chinese Academy of Sciences, Beijing 100094, China
2    School of Aeronautics and Astronautics, University of Chinese Academy of Sciences, Beijing 100049, China
3    Investigation College of People's Public Security University of China, Beijing 100038, China
4    China Aero Geophysical Survey and Remote Sensing Center for Land and Resources, Beijing 100083, China
*    Correspondence: cairong@aircas.ac.cn

**Abstract:** Few-shot hyperspectral classification is a challenging problem that involves obtaining effective spatial–spectral features in an unsupervised or semi-supervised manner. In recent years, as a result of the development of computer vision, deep learning techniques have demonstrated their superiority in tackling the problems of hyperspectral unmixing (HU) and classification. In this paper, we present a new semi-supervised pipeline for few-shot hyperspectral classification, where endmember abundance maps obtained by HU are treated as latent features for classification. A cube-based attention 3D convolutional autoencoder network (CACAE), is applied to extract spectral–spatial features. In addition, an attention approach is used to improve the accuracy of abundance estimation by extracting the diagnostic spectral features associated with the given endmember more effectively. The endmember abundance estimated by the proposed model outperforms other convolutional neural networks (CNNs) with respect to the root mean square error (RMSE) and abundance spectral angle distance (ASAD). Classification experiments are performed on real hyperspectral datasets and compared to several supervised and semi-supervised models. The experimental findings demonstrate that the proposed approach has promising potential for hyperspectral feature extraction and has better performance relative to CNN-based supervised classification under small-sample conditions.

**Keywords:** hyperspectral; unmixing; autoencoder; deep learning; few-shot; classification

## 1. Introduction

In earth observation, hyperspectral technology is one of the top trends in the remote sensing community, and it plays a significant role. The hundreds of spectral bands provided by hyperspectral images (HSIs) give them an inherent advantage in quantitative applications such as mineral mapping, environmental monitoring and classification [1].

Earlier work mainly focused on discriminative spectral feature extraction, such as the spectral angle mapper (SAM) [2] and spectral information divergence (SID) [3]. These methods utilize extensive prior knowledge and typically require no sample. They do not, however, fully use the spatial features of HSIs. Later on, some supervised spectral classifiers, such as support vector machines (SVM) [4], random forest [5] and neural networks [6], were proposed to achieve a more accurate classification and have since gained widespread acceptance. Nonetheless, the curse of dimensionality is still a bottleneck for supervised classifiers. To resolve this concern, a large number of spectral–spatial joint feature extraction methods have been developed for HSI classification [7–11]. In addition, numerous band selection methods [12,13] and hand-crafted feature extraction methods combined with a learning strategy are proposed [8,14]. However, these shallow features still have limitations regarding more precise hyperspectral classification in complex scenes.

Thanks to advances in computer vision, deep learning technology has made significant strides in remote sensing applications in recent years. CNNs' potent learning capabilities

have enabled end-to-end supervised deep learning models to achieve highly competitive hyperspectral classification results [15–18] when large amounts of labeled data are available. The various deep leaning classification modes for HSIs are summarized in [19]. However, since hyperspectral classification is a few-shot problem in most cases, it is challenging to collect the large number of hand-crafted training samples required for supervised classification models. In addition, larger models applied to scenarios with limited data tend to result in overfitting and reduce the robustness of the model. Currently, developing an unsupervised or semi-supervised method suitable for small-sample scenarios is a significant challenge in hyperspectral classification.

Various deep learning-based approaches with different learning paradigms have been proposed to address this issue [20,21], including transfer learning [22,23], few-shot learning [24,25] and self-learning [26,27]. The purpose of transfer learning is to initialize the network weights, thereby reducing training time and improving accuracy. Few-shot learning also employs the strategy of transfer learning but focuses more on mining meaningful representation information from labeled data [28]. However, this method has stringent requirements for the labeled quality and diversity of the data. Unlike few-shot learning methods, self-supervised learning can learn deep representations by reconstructing the input completely. However, this type of data representation is regarded as the compression of all information rather than the effective screening of meaningful information, and information with a small proportion is often disregarded [29].

Summarizing previous research, we believe that in order to solve the problem of few-shot hyperspectral classification, three conditions are necessary: 1. Strong feature extraction capabilities; 2. Effective representation information is obtained from data; 3. The model is robust and less dependent on data. Therefore, the question of how to combine self-supervised learning with traditional physical models, such as HU, to achieve effective expression of representational information has become an important area of study.

Let us take a look at the HSI unmixing methods first. The HU technique is applied in order to split the mixed pixel spectrum into materials in their purest forms (endmembers) as well as their proportions (abundances). Generally, the common HU model can be summarized in two categories: the linear mixing (LM) model and the nonlinear mixing (NLM) model [30]. The LM model operates under the presumption that every pixel of the HSIs is a linear combination of the pure endmembers. These methods can be further subdivided into pure pixel-based methods [31,32], and non-pure pixel-based methods [33,34], which concentrate on leveraging the data structure by making geometrical or sparseness assumptions. However, the LM model does not consider the multiple-scattering function and the interaction between objects, which makes it unsuitable to solve the HU problem in complex scenes [35]. In recent years, many neural network (NN) algorithms have been proposed to handle NLM challenges [36–38]. Deep learning methods have been proven to improve the accuracy of HU. Recently, the convolutional autoencoder (CAE) has emerged as a new trend in HU applications [39–41]. A denoising and sparseness autoencoder is introduced in [42,43] to estimate the abundance of endmembers. Additionally, stacked autoencoders are further employed for HU [44,45]. Most recently, 3D-CNN autoencoders [35,46] were employed to handle HU problems in a supervised setting. Existing research has acknowledged the significance of spatial–spectral joint characteristics for classification [47]. New advances indicate that HU theory can guide the network to learn more effective and regularized representational features; thus, the robustness of the model can be enhanced. The endmember abundance obtained through HU can provide useful spatial–spectral features for semi-supervised classification [48–50]. This also sheds new light on solving the few-shot classification problem.

In this research, we introduce a novel end-to-end convolutional autoencoder that we call CACAE. Furthermore, an attention mechanism is utilized to increase the accuracy of abundance estimation by extracting the diagnostic spectral characteristics associated with a given endmember more precisely. In addition, a semi-supervised classification pipeline based on CACAE that uses endmember abundance maps as classification features

is introduced. Experiments are carried out on real hyperspectral datasets, and the outcomes are compared using a variety of supervised and semi-supervised models.

The remaining sections of this paper are structured as follows: In Section 2, the proposed method is described. The experimental dataset is described in Section 3. Experiments and analysis are discussed in Section 4, while the final section provides the conclusion.

## 2. Methods

### 2.1. Problem Definition

The LM model typically assumes that the reflectance spectra are linearly mixed by different endmembers. The HU formula can be expressed as:

$$M = EA + N \tag{1}$$

where $M$ represents a mixed pixel, $E$ stands for the endmembers, $A$ stands for the abundances and $N$ is the additive vector. However, considering the multiple-scattering function and the interaction between objects, the nonlinear model is more suitable to solve the HU problem in complex scenes. The nonlinear model is defined as:

$$M = g(EA) + N \tag{2}$$

where $g$ refers to a nonlinear function.

Autoencoders are unsupervised training networks that are designed to force the learned representations to exhibit useful properties. During self-learning, input data are compressed, and data structures are subsequently learned and utilized. The decoder can be viewed as the reconstruction of HSIs by endmembers and abundances. The abundances can be estimated via an autoencoder by providing the endmembers as the weight of the rebuilt layer. Furthermore, the stacked convolutional and activation layers of CNNs can be interpreted as a nonlinear transformation of the linear function.

### 2.2. Attention-Based 3D Autoencoder

The spectral dimension is redundant for hyperspectral data. Consequently, we employ 3D convolution to make the extraction of effective features more feasible. The formula is expressed as follows:

$$p_{lf}^{xyz} = \lambda \left( \sum_{n} \sum_{h=0}^{H_k-1} \sum_{w=0}^{W_k-1} \sum_{d=0}^{C_k-1} w_{lfn}^{hwd} p_{(l-1)n}^{(x+h)(y+w)(z+d)} \right) + b_{lf} \tag{3}$$

where $p_{lf}^{xyz}$ is the value at position $(x, y, z)$ on the $f$th feature map in the $l$th layer and $n$ is an index of the sets on the previous $(l-1)$ layer before that; $H_k$, $W_k$ and $C_k$ are the kernel's height, width and channel, while $w_{lfn}^{hwd}$ is the weight at position $(h, w, d)$ associated with the $f$th feature map; $b$ represents the bias and $\lambda$ stands for the activation function.

We use the hyperspectral cube as input to make sure that both the spatial and spectral information is considered when estimating the abundance (Figure 1). Assume that a hyperspectral cube has the dimensions S × S × C, where S represents the size of the spatial window and C represents the total number of spectral bands. The spatial and spectral information around the center pixel can be used to estimate the related abundance vector of the central pixel. Table 1 shows the architecture of the CACAE model. An encoder with an attention mechanism consisting of four 3D convolutional layers can further filter the effective channel information of HSIs. In order to implement the non-negativity and summation constraints [51,52] of HU, we add an NSC layer to the decoder part through the use of the softmax activation function.

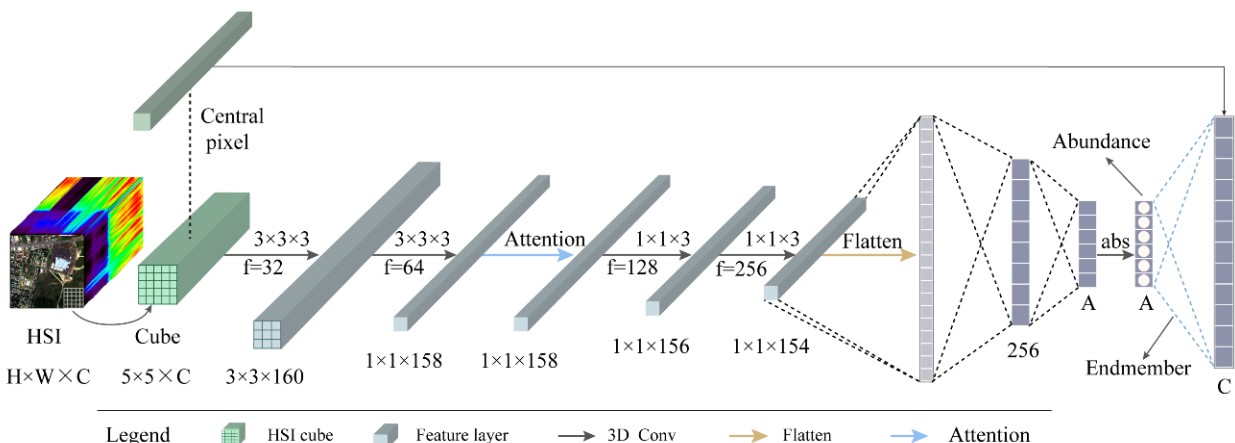

**Figure 1.** Diagram of the proposed 3D convolutional neural network.

**Table 1.** The structure of the CACAE model.

| Layers | Kernel | Filter | Activation | Feature |
|---|---|---|---|---|
| 3DConv | (3, 3, 8) | 32 | LeakRelu | (3, 3, C-7, 32) |
| 3DConv | (3, 3, 8) | 16 | LeakRelu | (1, 1, C-14, 16) |
| 3DConv | (1, 1, 8) | 8 | LeakRelu | (1, 1, C-21, 8) |
| 3DConv | (1, 1, 8) | 2 | LeakRelu | (1, 1, C-28, 2) |
| Attention | | | | |
| Flatten | - | - | - | (C-28) $\times$ 2 |
| Dense | - | 32 | LeakRelu | 32 |
| Dense | - | N | LeakRelu | N |
| NSC | - | - | softmax | N |
| Dense-3 | - | C | linear | C |

HSIs contain hundreds of spectral bands, but not all spectral information is useful. The attention mechanism performs the extraction of the diagnostic spectral features that are associated with the given endmember by reassigning the weights to the spectral band. Figure 2 illustrates the suggested module for the channel attention mechanism, which is described in this section.

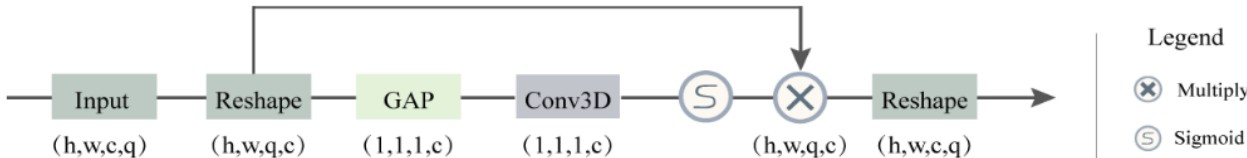

**Figure 2.** The architecture of the channel attention module.

In the first step, the dimensions of the input feature map must be transformed into the channel-last form. Then, global average polling is utilized in the process of integrating and compressing the data of each channel into a one-dimensional vector. The weighted layer, which represents the importance of the feature map of each channel, is created using a $1 \times 1$ convolutional layer and sigmoid activation.

$$s = g(z, W) = \lambda(W_2 \delta(W_1 z)) \tag{4}$$

in which $W_1 \in \mathbb{R}^{\frac{C}{r} \times C}$ and $W_2 \in \mathbb{R}^{C \times \frac{C}{r}}$. $\lambda$ and $\delta$ are activation functions.

$$\widetilde{x}_c = s_c F_c \tag{5}$$

where $\widetilde{X} = [\widetilde{x}_1, \widetilde{x}_2, \ldots, \widetilde{x}_C]$ represents the channel-wise multiplication of the vector $s_c$ and the feature map $F_c \in \mathbb{R}^{H \times W}$.

### 2.3. Loss Function

The spectral angle distance function (SADF) is employed for an abundance estimation process that deals with calculating the similarity between the rebuilt spectrum and the original spectrum. The SADF is defined as follows:

$$SADF = arccos\left(\frac{<\hat{V}_j, V_j>}{||\hat{V}_j||_2 ||V_j||_2}\right) \tag{6}$$

where $\hat{V}_j$ is the spectrum reconstructed by the model and the $V_j$ is the original input.

Cross-entropy is used in the classification process to measure the discrepancy between the predicted probability distribution and the actual probability distribution. It shows how far the actual output is from the expected output. This means that the two probability distributions are closer when the cross-entropy value is smaller. The equation can be written as:

$$L = \frac{1}{m}\sum_i L_i = -\frac{1}{N}\sum_i \sum_{c=1}^{k} y_{ic} log(p_{ic}) \tag{7}$$

where $m$ is the total number of the sample size, $k$ denotes the number of categories and $y_{ic}$ is an indicator variable that can take on the values 0 or 1 (it takes on the value 1 if the category $c$ is the same as the category of sample $i$ and it takes on the value 0 otherwise). Finally, $p_{ic}$ represents the predicted probability that the sample $i$ is a member of category $c$.

## 3. Materials and Experimental Setup

### 3.1. Dataset

To test the suggested method, it was applied to two publicly available datasets with GT. Following is a description of the datasets' specifics.

#### 3.1.1. Jasper Ridge Dataset

The AVIRIS sensor is responsible for the collection of the Jasper Ridge dataset, which is now one of the most popular hyperspectral datasets utilized in hyperspectral research. It features 224 bands ranging from 0.38 to 205 μm and a resolution of 100 by 100 pixels. There are some bands associated with atmospheric effects that were eliminated. Experimentation was conducted on the remaining 198 bands. The Jasper Ridge data include four endmembers, namely "Tree", "Water", "Soil" and "Road" (Figure 3).

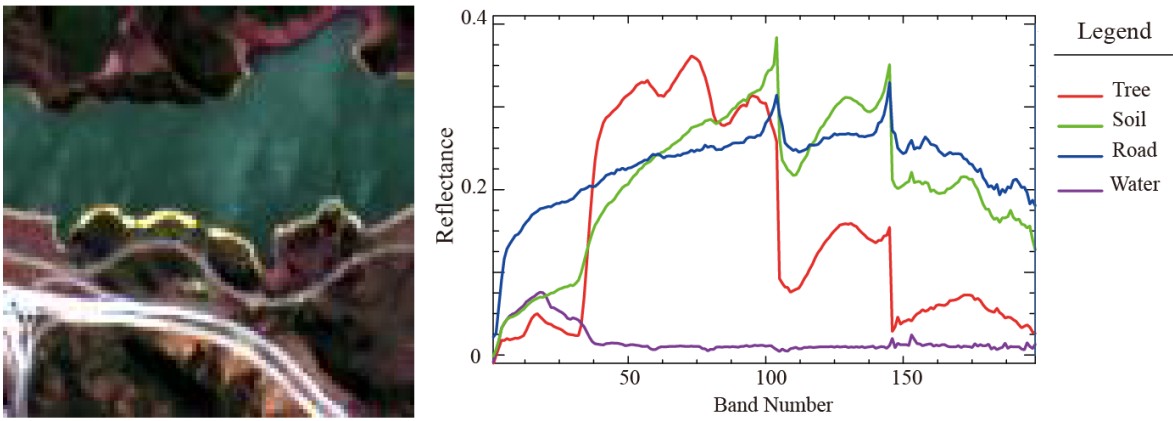

**Figure 3.** The Jasper Ridge data shown in RGB form and ground truth endmembers.

### 3.1.2. Urban Dataset

Additionally, the Urban dataset is a widely utilized HSI dataset for unmixing research. It consists of 307 by 307 pixels and 210 bands between 0.4 and 2.5 um. The 48 undesirable bands were eliminated, leaving 162 bands for experimentation. The data contain six endmembers, namely "Asphalt", "Grass", "Tree", "Roof" and "Dirt" (Figure 4).

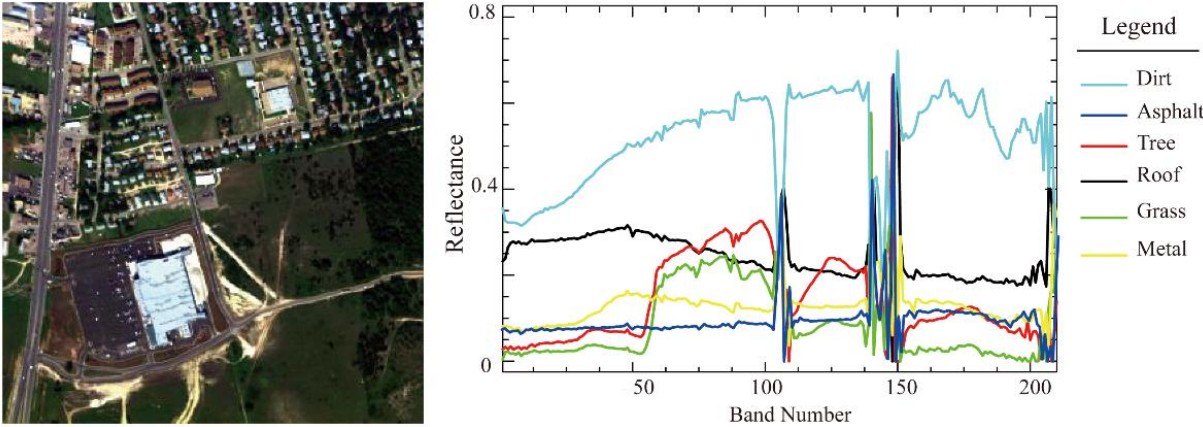

**Figure 4.** The Urban dataset shown in RGB format as well as GT endmembers.

### 3.1.3. Shenyang Dataset

The Shenyang dataset was captured in Shenyang, Liaoning Province, China, by a next-generation Chinese airborne hyperspectral sensor, the airborne multi-modular imaging spectrometer (AMMIS), which was developed by the Shanghai Institute of Technical Physics (SITP). The spatial resolution of this dataset is 0.75 m/pixel. In experiments, 190 spectral bands were used, and the wavelength range was 0.4–0.9 μm [53–55]. The dataset consists of 651 × 837 pixels and five different ground objects, including "Tree", "Grass", "Rice", "Corn" and "Bare land". GT endmembers were manually labeled with reference to the ground investigations. A visualization image and the spectral curves of the endmembers in the scene depicted are shown in Figure 5.

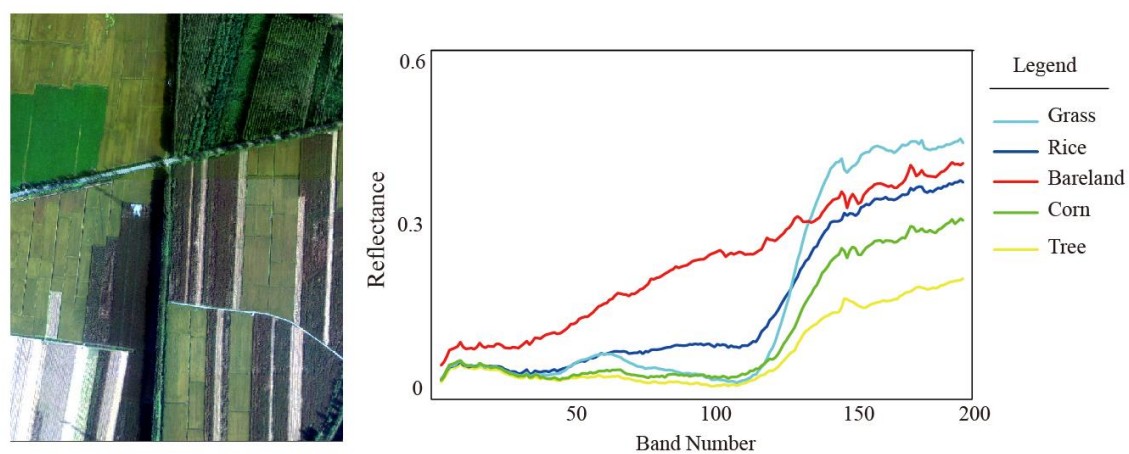

**Figure 5.** The Shenyang dataset shown in RGB format as well as GT endmembers.

### 3.2. Experiment Setting

In our experiments, all comparative CNN models used the same data and training parameter settings as the proposed methods. For Jasper and Urban HSIs datasets, only 10% of pixels were used for training, and all pixels of the datasets were used for evaluation. For the Shenyang dataset, we constructed 7 sets of training datasets through random sampling according to the sampling rates of 10%, 1%, 0.5%, 0.2%, 0.1% and 0.05%, which were used

to test the robustness of the algorithm under small-sample conditions. During training, experiments were run in the TensorFlow (2.9.1) environment on an NVIDIA Tesla A100 GPU resource supplied by the HeyWhale data mining platform [56] and optimized using an adaptive moment estimate technique with an initial learning rate of 0.0005. In order to avoid overfitting (despite the low number of training data samples), another approach called dropout with a rate of 0.2 was used. Training consisted of a total of 100 epochs, and the batch size was 30.

## 4. Experiments and Analysis

### 4.1. Comparison Model

To demonstrate the efficacy of the input data and the attention mechanism, we developed three additional 3DCAE-based models for comparison: PCAE, CCAE and PACAE. They utilize the same infrastructure as CACAE but differ in specific details. The characteristic features of CACAE are that its input is a three-dimensional image cube and it utilizes the attention module. The difference between PACAE and CACAE is that PACAE's input is a central pixel rather than an image cube. The main difference between PCAE, CCAE and CACAE is that the structures of PCAE and CCAE lack an attention module. Table 2 details the differences between the various models.

**Table 2.** Comparison table of the differences between all 3DCAE-based models.

| Model | Input Data | Attention | Basic Model |
|:---:|:---:|:---:|:---:|
| CACAE | Cube | Yes | 3D-CAE |
| PACAE | Pixel | Yes | 3D-CAE |
| PCAE | Pixel | No | 3D-CAE |
| CCAE | Cube | No | 3D-CAE |

In ground object classification, in addition to PACAE, the classic spectral feature analysis methods of SAM and SID are applied as comparative approaches. A traditional machine learning method for abundance estimation, fully constrained least squares (FCLS) [57], was also employed to assess the effectiveness of the recommended approach for feature extraction. In order to further assess the advantages and disadvantages of the suggested semi-supervised and supervised models, a basic CNN model with a fundamental architecture similar to that of CACAE is presented (Table 3).

**Table 3.** The structure of the basic CNN model.

| Layers | Kernel | Filter | Activation | Feature |
|:---:|:---:|:---:|:---:|:---:|
| Conv2D | (3, 3) | 32 | Relu | (3, 3, C-7) |
| Conv2D | (3, 3) | 16 | Relu | (1, 1, C-14) |
| Conv2D | (1, 1) | 8 | Relu | (1, 1, C-21) |
| Conv2D | (1, 1) | 2 | Relu | (1, 1, C-28) |
| Flatten | - | - | - | (C-28) $\times$ 2 |
| Dense | - | 32 | Relu | 32 |
| Dense | - | N | softmax | N |

### 4.2. Evaluation

Due to the fact that the method proposed in this paper involves two main processes, endmember abundance map estimation and hyperspectral classification, a quantitative evaluation of the two processes is necessary. As mentioned earlier, the higher the accuracy of endmember abundance map estimation, the more meaningful the extracted spatial–spectral features are for classification. The *RMSE* judges the similarity between the predicted value and the estimated value by measuring their absolute distance, whereas the *ASAD* evaluates from the perspective of spectral similarity, with a smaller value indicating a smaller difference. Both of the aforementioned metrics are appropriate for evaluating the

gap between the estimated endmember abundance and the ground truth (GT). *RMSE* and *ASAD* are defined as follows:

$$RMSE = \sqrt{\frac{\sum_{i \in N}(AP_i - AT_i)^2}{N}} \tag{8}$$

$$ASAD = cos^{-1}\left(\frac{T^T P}{\| T \| \| P \|}\right) \tag{9}$$

where $N$ denotes the dataset size, true abundance is represented by $T$, while predicted abundance is represented by $P$.

During the hyperspectral classification process, several regularly adopted classification accuracy evaluation criteria, including precision, recall, accuracy, *F*1-score and MIoU, are used to assess the comparative approaches. A comprehensive analysis of classification accuracy metrics is presented in [58]. These metrics have the following definitions:

$$P = \frac{TP}{TP + FP} \tag{10}$$

$$R = \frac{TP}{TP + FN} \tag{11}$$

$$OA = \frac{TP + TN}{TP + TN + FP + FN} \tag{12}$$

$$F = \frac{2}{\frac{1}{Recall} + \frac{1}{Precision}} \tag{13}$$

$$MIoU = \frac{1}{k+1} \sum_{i=0}^{k} \frac{TP}{FN + FP + TP} \tag{14}$$

where $P$, $R$, $F$, $OA$ and $MIoU$ represent precision, recall, *F*1-score, overall accuracy and mean intersection over union, respectively. $TP$, $TN$, $FP$ and $FN$ represent true positive, true negative, false positive and false negative, respectively, for each class $k$.

### 4.3. Abundance Map Estimation

In this section, the presented model is implemented to Jasper Ridge and Urban datasets for the purposes of quantitative and visual analysis. In this step, the estimated abundances of each endmember are compared to the corresponding GT. Several 3D-CNN autoencoders with similar structures are also employed to evaluate the abundance estimation performance of the proposed model. Then, classification applications based on abundance results are carried out, using multiple semi-supervised and supervised models for classification accuracy comparison.

#### 4.3.1. Abundance Map Estimation for Jasper Ridge Dataset

The estimated abundance and the reference GT abundance for the Jasper Ridge dataset are shown in Figure 6. At first glance, all algorithms work quite well in this task, indicating that the 3D-CNN-based autoencoder has great potential for HU. However, through careful comparison of the estimation error (EE) map, it is not difficult to find that the proposed method has a superior visual performance when compared to the method that was used for comparison.

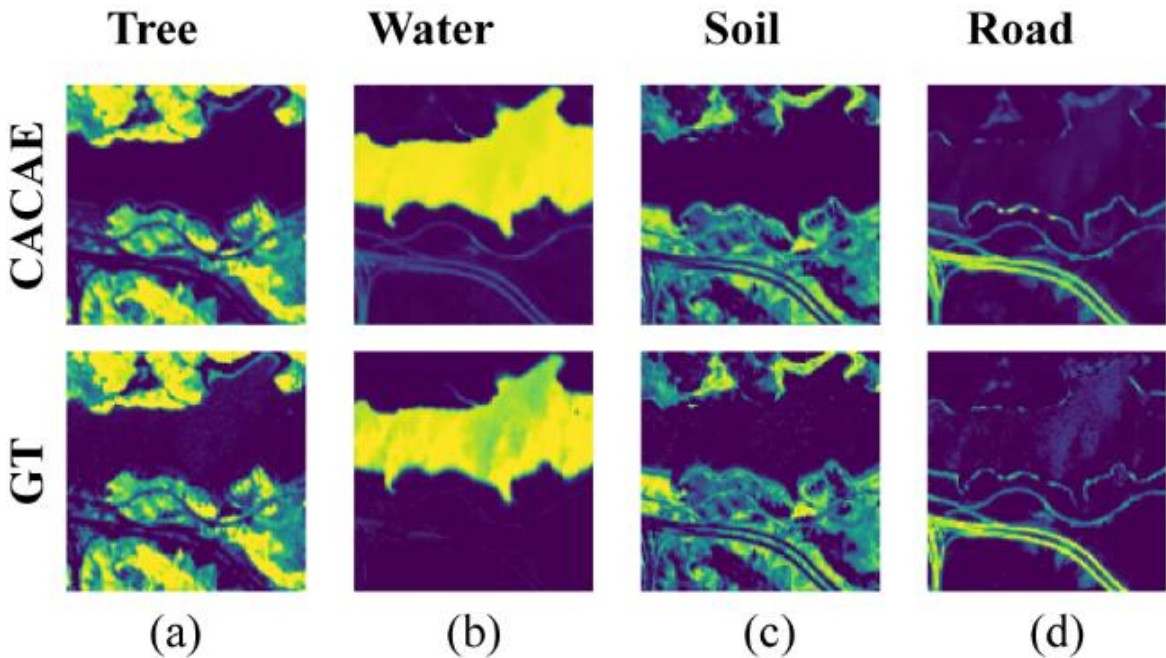

**Figure 6.** Abundance maps estimated by CACAE and the true abundance maps from the Jasper Ridge dataset. (**a**) Tree; (**b**) Water; (**c**) Soil; (**d**) Road.

Table 4, with the best results highlighted in bold, indicates that CACAE has higher prediction accuracy for most endmembers, with smaller RMSE and ASAD. For water objects, both CACAE and CCAE achieved good and close estimation results for abundance. It is worth noting that the performance of the model with HSI cube input is superior to that with central pixels as the input. In addition, under the assumption of the same input (cube or pixel), the performance of the attention-based model performs better in most cases, as evidenced by and highlighted in the Urban dataset.

**Table 4.** RMSE and ASAD results for all endmembers of the Jasper Ridge dataset ($\times 10^{-1}$).

| Dataset | Endmember | Algorithms | PCAE | CCAE | PACAE | CACAE |
|---------|-----------|------------|------|------|-------|-------|
| Jasper | Tree | RMSE | 0.468 | 0.422 | 0.539 | 0.419 |
| | | ASAD | 0.920 | 0.836 | 1.008 | 0.830 |
| | Water | RMSE | 1.146 | 0.831 | 0.975 | 0.835 |
| | | ASAD | 1.915 | 1.365 | 1.594 | 1.380 |
| | Dirt | RMSE | 1.137 | 0.874 | 1.051 | 0.792 |
| | | ASAD | 2.269 | 2.078 | 2.332 | 1.910 |
| | Road | RMSE | 0.908 | 0.818 | 0.763 | 0.741 |
| | | ASAD | 3.392 | 3.046 | 3.392 | 2.657 |
| | Sum | RMSE | 0.955 | 0.759 | 0.856 | 0.716 |
| | | ASAD | 2.240 | 1.773 | 2.004 | 1.671 |

### 4.3.2. Abundance Map Estimation for Urban Dataset

The Urban dataset, with more endmembers and a larger image size, is considered to provide a more objective assessment compared to the Jasper Ridge dataset. Figures 7 and 8 show the GT abundance and the estimated abundance, respectively, for the Urban dataset, for a straightforward visual comparison. When compared to the other 3D-CNN models, the experimental results show that CACAE achieves better performance, with lower EE and abundance estimation results that are most similar to the GT. As shown in Figure 8a,e,f, the performance enhancement achieved by the present work is clearly apparent on the Asphalt, road, Metal and Dirt EE maps.

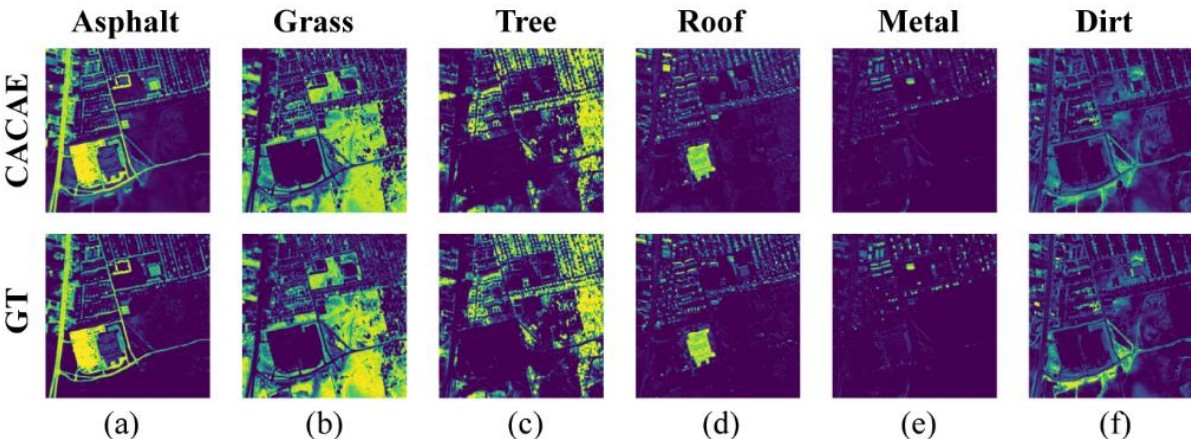

**Figure 7.** Abundance maps estimated by CACAE and the true abundance maps from the Urban dataset. (**a**) Asphalt; (**b**) Grass; (**c**) Tree; (**d**) Roof; (**e**) Metal; (**f**) Dirt.

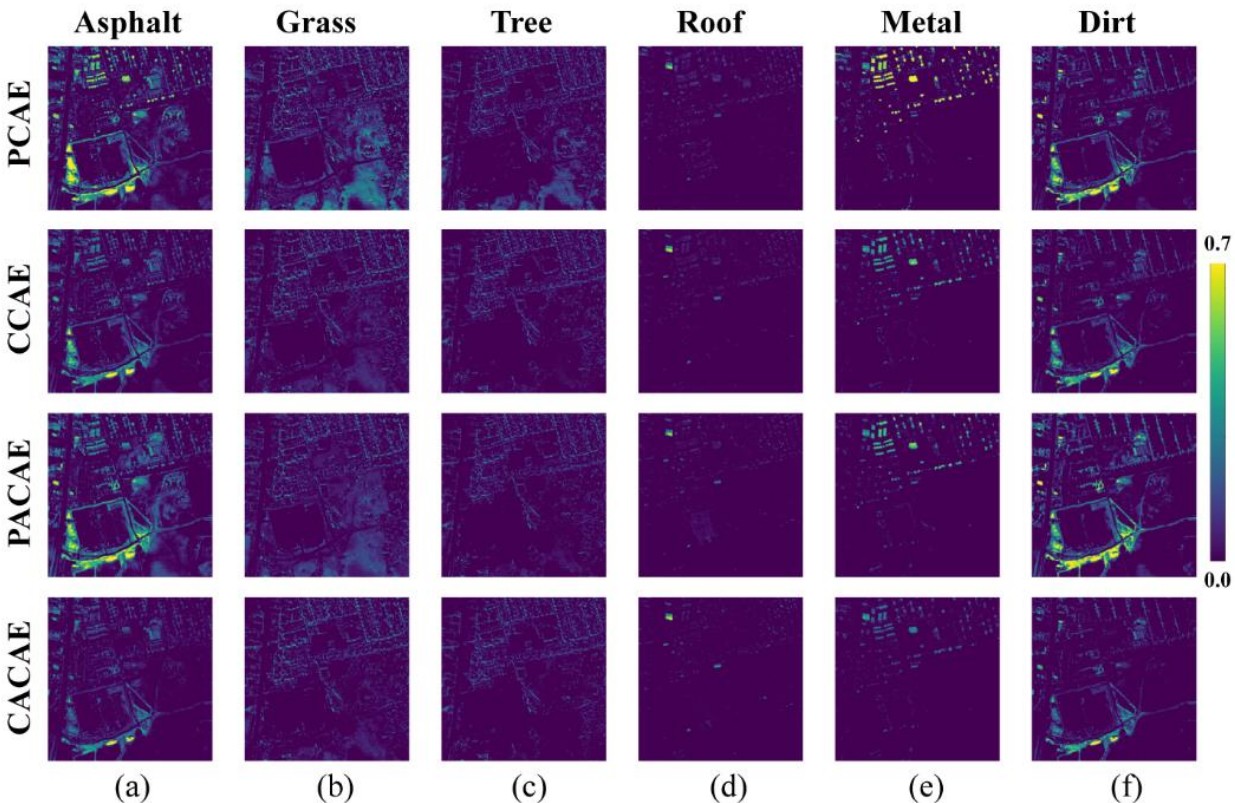

**Figure 8.** Absolute difference from the abundance of the Urban dataset estimated by different methods. (**a**) Asphalt; (**b**) Grass; (**c**) Tree; (**d**) Roof; (**e**) Metal; (**f**) Dirt.

Table 5 reports the RMSE and ASAD of each endmember for the Urban dataset. It shows that CACAE has the lowest estimation error according to the RMSE and ASAD values for all endmembers, with values of $0.83 \times 10^{-1}$ and $2.50 \times 10^{-1}$, respectively. Once again, the attention-based model still outperforms the others. Additionally, it is observed that the models that take the HIS cube as input lead to better results in most cases.

**Table 5.** RMSE and ASAD results for all endmembers of the Urban dataset ($\times 10^{-1}$).

| Dataset | Endmember | Algorithms | PCAE | CCAE | PACAE | CACAE |
|---|---|---|---|---|---|---|
| Urban | Road | RMSE | 1.694 | 1.243 | 1.667 | 1.017 |
| | | ASAD | 4.189 | 3.256 | 4.108 | 2.711 |
| | Grass | RMSE | 1.456 | 1.071 | 1.136 | 0.981 |
| | | ASAD | 2.805 | 2.162 | 2.183 | 1.980 |
| | Tree | RMSE | 1.038 | 0.881 | 0.872 | 0.833 |
| | | ASAD | 2.482 | 2.123 | 2.099 | 2.006 |
| | Roof | RMSE | 0.538 | 0.469 | 0.459 | 0.450 |
| | | ASAD | 2.463 | 2.171 | 2.143 | 2.100 |
| | Metal | RMSE | 1.166 | 0.743 | 0.753 | 0.536 |
| | | ASAD | 9.485 | 5.575 | 6.002 | 3.780 |
| | Dirt | RMSE | 1.347 | 1.117 | 1.569 | 0.977 |
| | | ASAD | 4.538 | 3.361 | 3.771 | 2.760 |
| | Sum | RMSE | 1.260 | 0.956 | 1.160 | 0.830 |
| | | ASAD | 3.850 | 2.893 | 3.532 | 2.503 |

### 4.4. Classification

In this section, classification applications based on abundance results are carried out, using multiple semi-supervised and supervised models for classification accuracy comparison.

#### 4.4.1. Ground Object Classification for Urban Dataset

Feature extraction is crucial for hyperspectral analysis. Over the past few years, as a result of the practical learning capabilities of CNNs, end-to-end supervised deep learning models have shown very competitive results in hyperspectral classification. However, since hyperspectral classification is a small-sample problem in most cases, the application of supervised classification is more labor-intensive and prone to overfitting. The endmember abundance obtained through HU has become an effective spatial–spectral feature source for semi-supervised classification. The semi-supervised classification process recommended in this paper is shown in Figure 9.

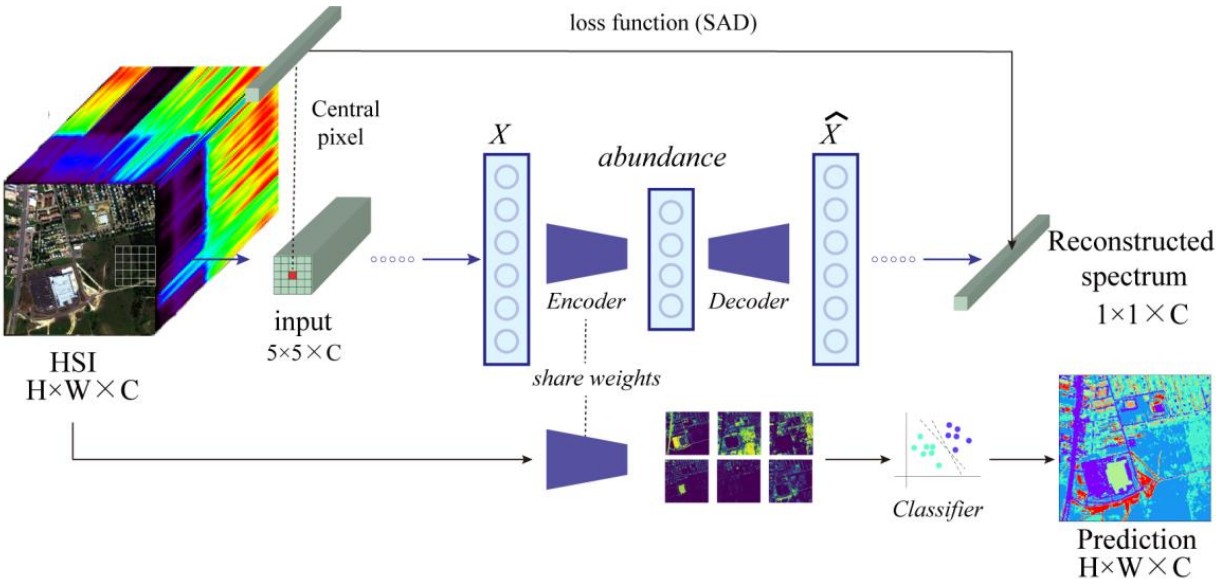

**Figure 9.** An overview of the proposed 3D autoencoder-based semi-supervised classification framework.

Based on the encoder of CACAE, trained through self-supervised learning, the Urban dataset is predicted to obtain its abundance estimation map. Then, we use a traditional

classifier to calculate the maximum abundance in each abundance feature category as the label of the classification result. Comparative experiments are performed on the classic semi-supervised and supervised methods (Figure 10). The figure demonstrates that the CNN-based method has visual performance comparable to the conventional method but is improved greatly. There are obvious misclassifications in the SID results. FCLS has poor classification performance due to inaccurate abundance estimation. SAM is slightly better than the former two methods but misclassifies the Soil and Metal categories more seriously. It is clear from careful observation that the visual effects of PACAE and CACAE are more accurate than the basic CNN model.

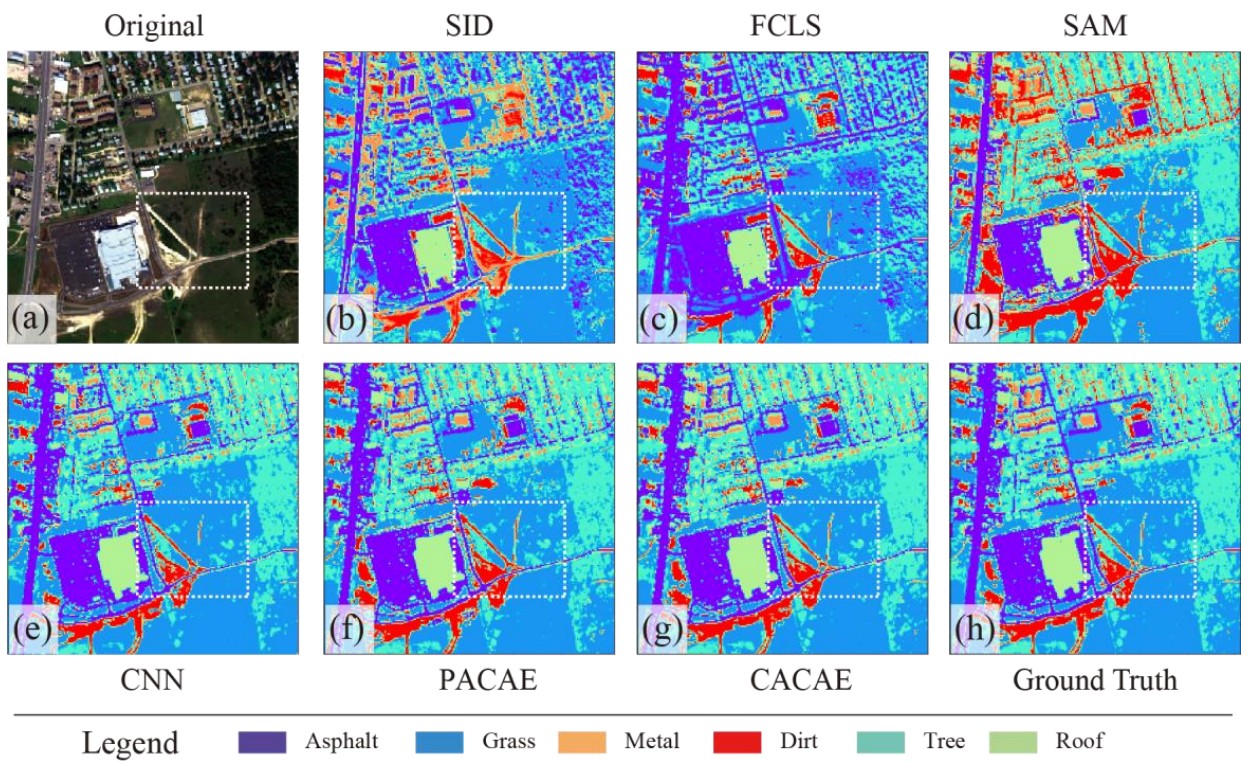

**Figure 10.** The classification results for the Urban dataset by different methods. (**a**) Original HSI in RGB format; (**b**) SID; (**c**) FCLS; (**d**) SAM; (**e**) CNN; (**f**) PACAE; (**g**) CACAE; (**h**) Ground Truth.

Table 6 records the quantitative evaluation results of all six models. The results reveal that the proposed method CACAE achieves the best outcomes across all metrics, and is slightly higher than the PACAE model, indicating that using image cubes as input data can provide rich spectral–spatial information that is more conductive to abundance estimation and classification. It is noticed that, in the case of 1/10 sample data, the accuracy of the proposed semi-supervised classification methods outperforms that of the CNN-based supervised classification model. Furthermore, the MIoU metrics have improved by nearly 7%, indicating that the proposed method has better performance under the conditions of relatively sufficient samples.

**Table 6.** Quantitative analysis of prediction results on the Urban dataset.

| Methods | Precision | Recall | OA | MIoU | F1-Score |
|---------|-----------|--------|------|------|----------|
| SID | 0.51 | 0.67 | 0.63 | 0.40 | 0.54 |
| FCLS | 0.62 | 0.80 | 0.75 | 0.51 | 0.66 |
| SAM | 0.85 | 0.81 | 0.83 | 0.68 | 0.80 |
| CNN | 0.84 | 0.86 | 0.90 | 0.75 | 0.85 |
| PACAE | 0.89 | 0.88 | 0.90 | 0.80 | 0.88 |
| CACAE | 0.91 | 0.89 | 0.92 | 0.82 | 0.90 |

### 4.4.2. Ground Object Classification for Shenyang Dataset

To further analyze the benefits of the proposed method for few-shot hyperspectral classification, we employ a real hyperspectral dataset from Shenyang, acquired by the airborne hyperspectral sensor AMMIS, for testing purposes. In the experiment, the classification performances of supervised classification methods (Basic-CNN and SVM) and semi-supervised classification methods (CACAE and PACAE) were compared at various sampling rates. In the supervised classification method, the performance of Basic-CNN is superior to that of the traditional machine learning method SVM, particularly under the condition of relatively sufficient samples, but both are controlled by the sample size, which is consistent with our prior knowledge (Figure 11). The superior performance of CACAE and PACAE proposed in this paper (Table 7) validates the earlier conclusion. In addition, their performance did not degrade significantly as the sampling rate decreased, suggesting that the proposed method is more robust. The MIoU value of the CACAE drops by only 1.7% when the sampling rate is decreased from 1/10 to 1/2000.

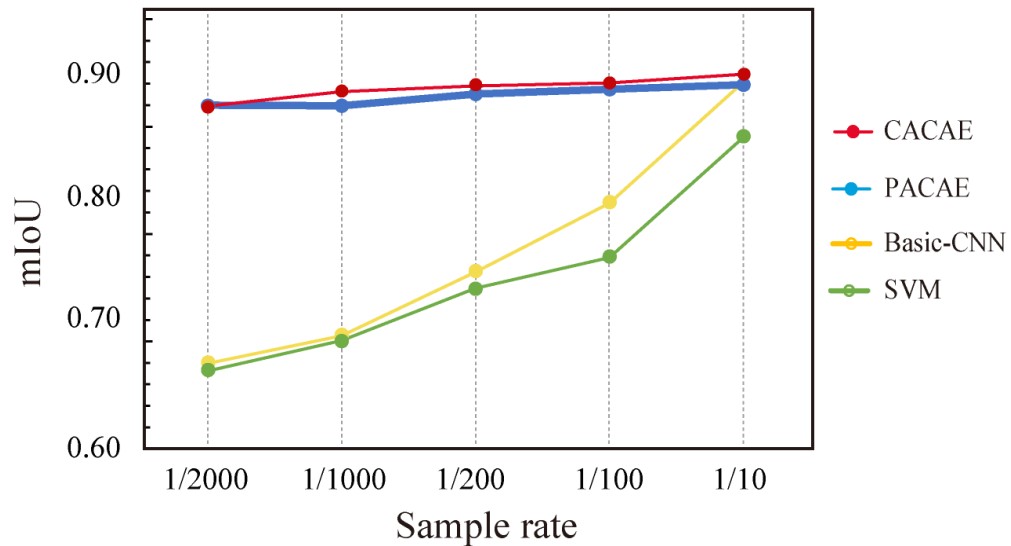

**Figure 11.** The trend of MIoU at different sampling rates.

**Table 7.** Comparison table of MIoU predicted by all models at different sampling rates.

| Methods | MIoU at Different Sample Rates | | | | | | | Mean |
|---|---|---|---|---|---|---|---|---|
| | 1/2000 | 1/1000 | 1/500 | 1/200 | 1/100 | 1/50 | 1/10 | |
| SVM | 0.659 | 0.683 | 0.701 | 0.726 | 0.751 | 0.801 | 0.848 | 0.752 |
| Basic-CNN | 0.665 | 0.687 | 0.750 | 0.739 | 0.795 | 0.843 | 0.893 | 0.784 |
| PACAE | 0.874 | 0.873 | 0.880 | 0.883 | 0.887 | 0.890 | 0.890 | 0.884 |
| CACAE | 0.873 | 0.885 | 0.882 | 0.889 | 0.892 | 0.892 | 0.899 | 0.890 |

Figure 12 illustrates the classification outcomes predicted by each of the four methods at varying sampling rates. It is evident that the prediction results of CACAE and PACAE are essentially unaffected by the sampling rate, and that the overall results are relatively consistent with the GT. With a higher sampling rate, the supervised classification method is more competitive, but as the sample size decreases, errors and omissions gradually appear, resulting in a decline in classification performance. Since CACAE uses more spatial information as input, the classification results are smoother and less noisy than those of other methods.

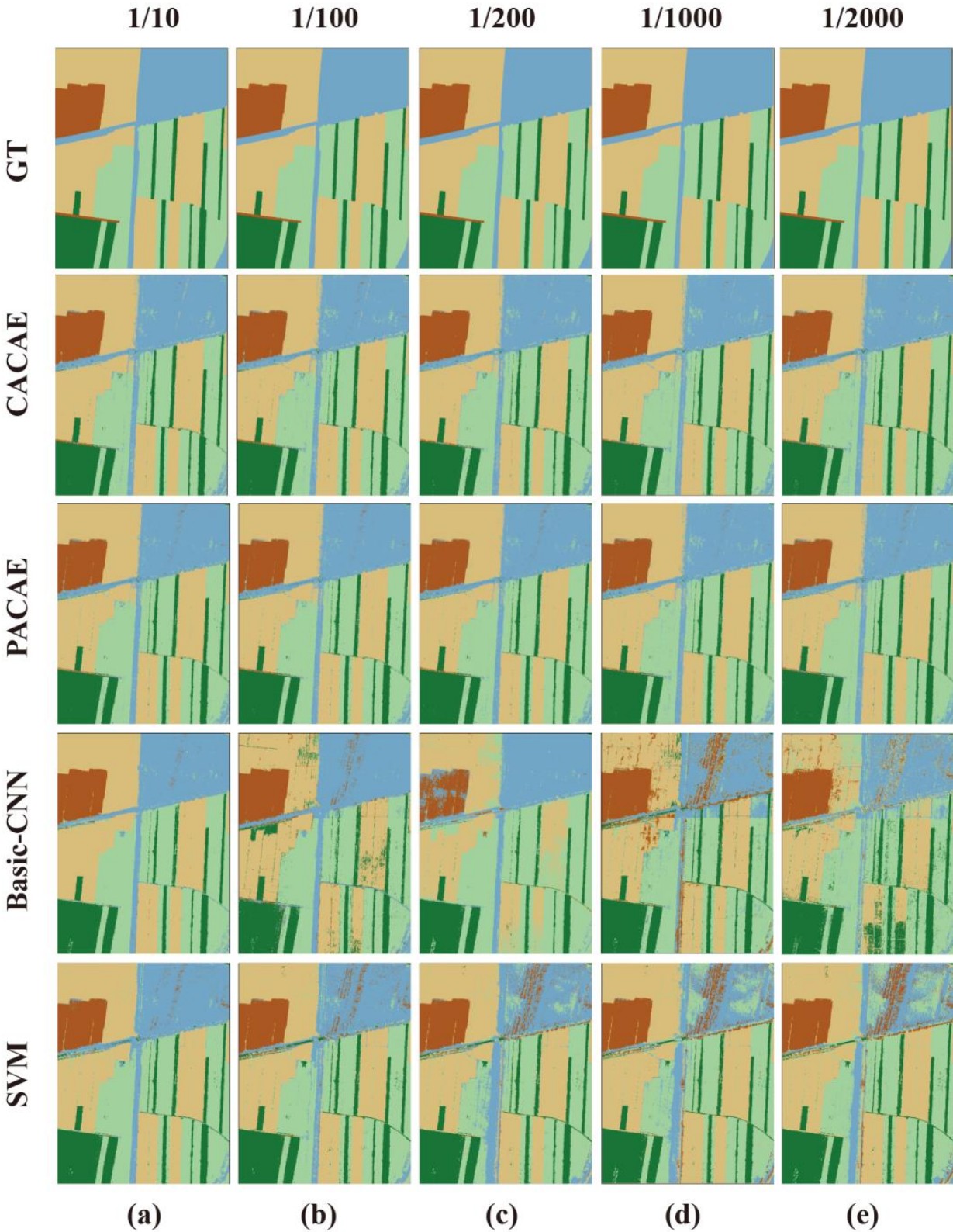

**Figure 12.** The classification prediction of all models at different sampling rates. (**a**) Sampling rate 1:10; (**b**) Sampling rate 1:100; (**c**) Sampling rate 1:200; (**d**) Sampling rate 1:1000; (**e**) Sampling rate 1:2000.

## 5. Conclusions

In this contribution, we present a new semi-supervised pipeline for few-shot hyperspectral classification, where endmember abundance maps obtained by HU are treated as latent features for classification. It includes two main processes. First, it uses 3D-CNN to build a self-supervised learning model and realizes the non-negativity and summation constraints through the NSC layer and realizes the extraction of endmember abundance information with given endmembers. Secondly, the abundance map can be treated as a diagnostic spatial–spectral feature, enabling claffication with only a small number of samples.

The first experiment is designed to verify the effectiveness of the proposed model for abundance map extraction. In this experiment, the performance of CACAE and other 3D-CNN methods (including PCAE, CCAE and PACAE) for HU is assessed (Section 4.3). The results suggest that the proposed model is capable of accurately estimating the abundance map of a given endmember with the lowest RMSE and ASAD. Additionally, the two strategies of utilizing the attention mechanism and taking an image cube as input are useful for improving the estimation accuracy. The second experiment tests the performance of proposed methods and other algorithms (including SID, FCLS, SAM, Basic-CNN and PACAE) for HSI classification (Section 4.4.1). The experimental results demonstrated that the performance of the proposed model not only outperformed traditional unsupervised and semi-supervised classification methods but also exceeded that of CNN-based supervised classification models, even when samples were relatively sufficient. It also illustrates the effectiveness of the abundance maps for HSI classification. The last experiment is designed to assess the robustness of the proposed model with a small sample size. The traditional machine learning algorithms SVM and Basic-CNN are employed as comparison models (Section 4.4.2). The proposed model shows advantages over the supervised classification model at all levels of sampling rates and shows stable and robust characteristics such that the model's performance is not restricted by the number of samples. These three experiments fully prove the potential of the proposed method in few-shot classification.

The advantage of the proposed method is that it combines a 3D-CNN with HU theory and uses the estimated endmember abundance as the input diagnostic feature for classification, which effectively improves the classification accuracy and robustness of the model. Because sufficient data are required to train a model with a large number of parameters, which tends to result in overfitting problems with small sample sizes, the suggested model employs a lightweight architecture design. The CNN model used for comparison also uses a similar backbone structure to make the comparison more fair and to avoid the difference in network structure affecting the reliability of the experiment.

The primary limitation of this method is that it is still dependent on the precision of the given endmembers. Although there are methods such as N-FINDR and VCA that can estimate endmembers, there is still room for improvement in accuracy. On the other hand, however, the current loss design makes it difficult to distinguish targets with very small spectral differences. Future research will focus on utilizing the autoencoder network for unsupervised endmember abundance estimation and developing a new loss function for more accurate classification. The hyperspectral datasets used in this paper and example code are available in the GitHub repository via https://github.com/lichunyu123/3DCAE_Hyper (accessed on 1 December 2022).

**Author Contributions:** Conceptualization, C.L.; methodology, C.L. and J.Y.; article writing and figure drawing, C.L.; writing—review and editing, C.L., R.C. and J.Y.; sample collection, C.L. and J.Y. All authors have read and agreed to the published version of the manuscript.

**Funding:** This research was funded by National Key Research and Development Program of China (2021YFC3000400).

**Acknowledgments:** The authors would like to thank Shanghai HeyWhale information Technology Company for providing online GPU computing resources.

**Conflicts of Interest:** The authors declare no conflict of interest.

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
