# Peer review of "An Attention-Based 3D Convolutional Autoencoder for Few-Shot Hyperspectral Unmixing and Classification"

_remotesensing, doi:10.3390/rs15020451_

Round 1
Reviewer 1 Report
The paper was well written and i have no comments for the same
Author Response
Response to Reviewer 1 Comments:
Common:
The paper was well written and i have no comments for the same
Response:
Thank you for your positive comments on our manuscript.
Reviewer 2 Report
I liked the paper and the presented experiments.
One low thing about the manuscript is the disappointingly short conclusion section.
Could you comment on other possible implementations or provide some comparison of the results and other existing methods for the used dataset?
The tables and images are misaligned in the manuscript - they should be reformatted.
The paper really misses the section with all used abbreviations list.
1. What is the main question addressed by the research?
The main topic addressed in the paper is the currently popular attention-based convolutional neural network. They are applied for a small set of hyperspectral images as an example application/check.
2. What does it add to the subject area compared with other published material?
The novelty of this is limited but could be an interesting example for some readers. To be honest, I cannot find some very unique things in the described application, but it uses a "fresh" and currently popular approach to such data.
3. What specific improvements should the authors consider regarding the
methodology? What further controls should be considered?
The abbreviations, like CACAE and CCAE should be carefully defined and commented on, this may not be obvious to all readers what stand for.
Similarly, with ASAD and RMSE, what is their definition, and secondly, what data in what way is used for this specific case to compute them? The paper misses that part.
4. Are the conclusions consistent with the evidence and arguments presented and do they address the main question posed?
The presented approach could be extended with additional checks with some other hyperspectral data, providing the information if this application works only for that small HSI set or have some more universal functionality.
5. Are the references appropriate?
- Good enough
6. Please include any additional comments on the tables and figures.
- Tables are wrongly enumerated (i.e., two tables 5)
- the information on how the results presented in the tables were computed have to be supplemented - on what part of the dataset? how many samples/pixels were used? what was this evaluation set structure?
- What was the difference between the evaluation and training set?
- where is this data accessible? Is there any way to verify this result? The authors do not say that.
Reviewer 3 Report
In this paper, we present a new semi-supervised pipeline for few-shot hyperspectral classification, where endmember abundance maps obtained by HU are treated as latent features for classification. In my opinion, the manuscript’s contributions are that the authors introduce a novel end-to-end convolutional autoencoder that called CACAE. However, some weaknesses should be addressed, especially the introduction and experiment.
Major issues:
1) In the introduction part, the current literature analysis is not enough to fully explain the significance of this study. For example, the popular few-shot hyperspectral classification and unmixing in recent years have not been given in detail. I suggest that the authors combine the introduction with relevant work and supplement relevant literature, such as
[1] Super-Resolution Mapping Based on Spatial-Spectral Correlation for Spectral Imagery [J]. IEEE Transactions on Geoscience and Remote Sensing, 2021, 59(3): 2256-2268.
[2] Target-Constrained Interference-Minimized Band Selection for Hyperspectral Target Detection, IEEE Transactions on Geoscience and Remote Sensing, 2021, 59(7): 6044-6064.
2) The description of the main contributions of this paper is not prominent enough. What problems does the network proposed by the authors mainly solve? Why must this problem be solved? The authors are suggested to add it at the end of the introduction.
3) The main structure of this paper is somewhat confused. It is suggested that the authors should put Section 2.3 Comparison Model and Section 2.5 Evaluation into the experiment section. And I also suggest that the data in Section 3 should not be a separate chapter, it should be put into the experiment section.
4) The experimental part needs further improvement. First, the selected comparison methods belong to a class of deep learning networks, and it is suggested that the authors compare the SOAT deep learning networks. In addition, traditional machine learning methods should be used as comparison methods. Finally, the selected experimental data are few, and at least one real data should be added for verification.
Minor issues:
1) There are some grammatical errors in the article, which need further careful proofreading. It is suggested that the authors invite professional personnel to proofread.
2) It is suggested that the authors analyze the advantages and disadvantages of the proposed model and its future prospects in the conclusion part of this paper.
Round 2
Reviewer 3 Report
Thank the authors for reply. I have no other questions.